# The Effect of Collectivism on Mental Health during COVID-19: A Moderated Mediation Model

**DOI:** 10.3390/ijerph192315570

**Published:** 2022-11-23

**Authors:** Yixuan Gao, Wenjie Yao, Yi Guo, Zongqing Liao

**Affiliations:** College of Psychology, Sichuan Normal University, Chengdu 610000, China

**Keywords:** COVID-19, mental health, expressive suppression, ego identity

## Abstract

Background: COVID-19 is an unprecedented public health emergency of international concern and has caused people to live in constant fear and posed a significant threat to their physical and mental health. Method: The study constructed a moderated mediation model to examine the mediating role of emotion regulation between collectivism and mental health and the moderating role of ego identity in the context of COVID-19. A total of 459 participants were recruited to complete the survey from 30 January to 8 May 2021.The Mental Health in COVID-19 Period Scale, Collectivism Tendency Scale, ERQ, and Identity Status Scale were used for the study. Results: (1) Expressive suppression played a mediating role in the relationship between collectivism and mental health; (2) The direct effect of collectivism on mental health and the path from expressive suppression to mental health were moderated by ego identity. Conclusion: The effect of collectivism on mental health is indirectly generated through expressive suppression and ego identity showing different patterns of regulation of mental health in different pathways, and its mechanisms and other important influences could be further explored in the future.

## 1. Introduction

The 2019 coronavirus (COVID-19) epidemic was declared to constitute a “public health emergency of international concern (PHEIC)” and has spread all over the world in the past two years [1]. By 6 November 2021, a total of 97,660 confirmed cases and 4636 deaths were reported nationwide (excluding Hong Kong, Macao, and Taiwan). Since December 2019, COVID-19 prevention and control measures have been enhanced seriously and the closure of cities, districts, and schools has become the norm. Severe prevention and control measures have hindered the spread of the virus to a certain extent but have caused anxiety and fear in people’s psyches. Coupled with the high contagiousness of COVID-19 and the lack of effective treatment, people live in fear of death, which seriously threatens people’s physical and mental health [2].

### 1.1. The Impact of Cultural Differences on Mental Health

The rapid spread of COVID-19 has a negative effect not only on individuals’ physical health, but also on their mental health. Many people showed various degrees of anxiety, fear, and worry [3]. They believed that COVID-19 would harm their mental health [4]. Furthermore, cultural differences should be considered when investigating the influence of the virus on daily life. Previous research showed that cultural values were buffers to psychological problems. For example, research has shown that collectivism could reduce stress perceptions by perceiving social support during COVID-19 [5]. Collectivism enhances psychological well-being by reducing anxiety and depression [6]. Even in some individualistic cultures, a higher collectivism level indicates a lower rate of depression [7]. However, some opposing research has revealed that collectivism leads to worse mental health. For example, researchers found that collectivism was positively associated with neuroticism, depression, and deregulation of emotions in a Russian sample [8]. It suggested that individualism rather than collectivism was a protective factor for depression in Russia, and collectivism promotes maladaptive emotional responses in emotionally unstable individuals in the face of environmental stress, ultimately leading to the emergence of depressive symptoms. Due to an excessive focus on relationships with others, people in collectivist cultures maintain close social relationships and are more focused on sharing and exchanging information with those around them, which leads to an increase in individual and collective perceived risk [9]. Hence, the impact of collectivism on mental health has not been conclusively established. 

Collectivists are interdependent and attentive to others’ feelings [10]. Furthermore, they like to share and collaborate with others [11]. Based on this, we hypothesized that, during the prevalence of COVID-19, people would share information they learn about COVID-19 with each other and would also be concerned about how those around them are affected by the epidemic (e.g., quarantine, death). Thus, this study hypothesized that collectivism during COVID-19 could influence people’s mental health.

### 1.2. The Mediating Role of Expressive Suppression

During COVID-19, we experienced various shocks, such as the illness of a friend and the death of a relative. Such bad news affects our mood and mental health. The use of emotion regulation has been proven to protect individuals’ mental or physical health, or both [12]. Previous research has suggested individuals mainly use cognitive reappraisal and expression inhibition to deal with bad emotions. People regulate their emotions by changing their evaluation of an object or event, which can be referred to as cognitive reappraisal. In addition, people inhibit their emotion-expressing behaviors when emotions are evoked, which can be referred to as expression inhibition. [13]. Emotion regulation has important implications for mental health, and adaptive emotion regulation has positive effects, such as diminishing negative emotions and maintaining psychological resilience and growth in response to stressful life events [14]. It has been found that cultural context can alter the form, frequency, and function of emotion regulation strategies [15]. In collectivist cultures, such as those in East Asia, it is relatively more encouraged to repress emotions than to express them [15]; that is, people in collectivist cultures prefer to express suppression.

There have not been consistent academic findings on expression inhibition and mental health. Some studies have found positive effects of expressive suppression, for example, a study with Chinese adolescents found that, when they chose attenuated regulation (including evaluative neglect and expressive inhibition) to manage and control their emotions, they were able to effectively reduce the impact of negative events on themselves [16]. Zhou and Wang [17] found that the inhibition of anger expression was negatively associated with depression in the emotional relationship with the mother. Collectivism also implies restraint in the expression of emotions and the non-public and indirect expression of personal feelings as a way to ensure group harmony [18,19]. Emotional repression, as a double-edged sword, is associated with negative mental health consequences [20,21,22]. Hu and his colleagues [22] conducted a meta-analysis of the relationship between emotion regulation and psychological well-being, finding that expression inhibition was negatively related to life satisfaction and positive mood. In addition, expressive suppression was also negatively related to psychological resilience and social adaptation [23,24]. Therefore, the present study hypothesized that collectivism would influence mental health through expressive suppression. 

### 1.3. The Moderating Role of Ego Identity

Erikson [25] argues that ego identity emphasizes the continuity of experience from the past and the present to the future. It begins in infancy, established in adolescence, while old age is a review and integration of the self. Ego identity is essential to people’s development and is a manifestation of good mental health, healthy personality development, and self-harmony [26]. Kato [27] divided ego identity into commitment, exploration, and crisis. Commitment refers to adherence to a set of convictions, goals, and beliefs. Exploration refers to the degree that individuals engage in a personalized search for different values, beliefs, and goals. Crisis refers to experiences where initial commitments are challenged [28]. Previous research found that commitment and exploration were positive predictors of mental health, but there was no consensus as to whether crisis had an impact on mental health [29]. A study by Chen [30] on high school and college student populations found that both commitment and exploration were positively related to mental health, while crisis was not significantly related to mental health. A study by Chen and Wang [31] further found that commitment and exploration negatively predicted anxiety, commitment negatively predicted depression, and crisis was not significantly related to anxiety and depression. Wei et al. [32] similarly found that commitment and exploration negatively predicted participants’ scores of all SCL-90 subscales (i.e., somatization, obsessive-compulsive, interpersonal sensibility, etc.), while crisis positively predicted depression and paranoid ideation. The results of crisis between Chen [30], Chen and Wang [31], and Wei et al. [32] are definitely different. Through contrasting the three studies in samples, scales, and other aspects, the gender breakdown was disproportional in Wei et al.’s [32] study. There were more male participants than female participants. We infer that the different results in crisis could be attributed to the disproportion of genders. All these studies demonstrated that the subscales of ego identity development are closely related to mental health. The present study examines how ego identity moderates people’s mental health in the context of COVID-19.

In summary, the present study constructs a moderated mediation model (Figure 1) to explore the three questions: (1) how collectivism affects people’s mental health during COVID-19; (2) how expressive suppression plays a mediating role between collectivism and mental health; and (3) how ego identity moderates people’s mental health in the context of COVID-19.

## 2. Materials and Methods

### 2.1. Participants

The present study recruited participants from Mainland China, using a convenience sampling method. The data were collected by administering an anonymous electronic questionnaire on the WenJuanXing public online platform (https://www.wjx.cn), and 629 participants completed the questionnaire from 30 January to 8 May 2021.

The invalid data were eliminated according to the following criteria: 1. Respondents completed the survey in less than 60 s; 2. Answers to all items were similar; 3. Answers to tests (e.g., “please choose 3 points”) were wrong. Overall, 170 invalid questionnaires were excluded, leaving 459 valid questionnaires for further analysis. Among the participants, the age range was from 15 to 60 and 36.6% were male, with an average age of 20.92 years (SD = 5.29).

### 2.2. Measures

#### 2.2.1. Mental Health during COVID-19 

The COVID-19 Mental Health Self-Assessment Scale adapted by Yu [33] was used to measure mental health during the COVID-19 epidemic. The scale includes four subscales: anxiety, fear, depression, and anger. The statements were assessed on a five-point Likert scale that ranged from 1 (not like me) to 5 (exactly me). According to the composite score, a higher score indicates fewer psychological problems and a higher level of mental health. Cronbach’s alpha for the present sample was 0.891. The Kolmogorov–Smirnov test showed that the data meet the condition of normal distribution: the skewness = 0.04, kurtosis = 0.19, D = 0.03, and *p*-value = 0.16 > 0.05.

#### 2.2.2. Collectivism Tendency

The scale we used is the proven scale used by Hooft and Jong [34]. Collectivism is measured directly on the individual level, with eight items measuring the collectivism tendency, such as “I often state my opinion directly and clearly” and “I prefer to perform a technical task in a group rather than individually”. Each item is rated from 1 (not like me at all) to 6 (exactly me). Cronbach’s alpha for the present sample was 0.726.

#### 2.2.3. Gross Emotional Regulation Questionnaire (ERQ)

The scale we used was developed by Gross and revised by Wang et al. [35], which has good reliability and validity in Chinese culture. It has 10 items, and the scale has two subscales: cognitive reappraisal (six items) and expressive suppression (four items). Items were rated from 1 (strongly disagree) to 6 (strongly agree). Cronbach’s alpha for the present sample was 0.780.

#### 2.2.4. The Identity Status Scale (ISS)

The Identity Status Scale [27] was used to measure identity commitment, exploration, and crisis. This scale was composed of three subscales: (a) Commitment (four items; e.g., “I know who I am and what I wish to do,”); (b) Exploration (four items; e.g., “I explore eagerly whatever I can engage in with all my might,”); (c) Crisis (four items; e.g., “I felt lost and had to reconsider seriously who I was and what I wanted to do,”). The statements were assessed on a Likert five-point scale from 1 “not like me at all” to 5 “exactly me”. Cronbach’s alpha for the present sample was 0.672.

### 2.3. Data Analysis

IBM SPSS Statistics for Windows, Version 26.0 (IBM, Chicago, IL, USA), and PROCESS Macro (Andrew F. Hayes, Calgary, AB, Canada) was utilized for data analyses. The specific procedure of data analysis was as follows: descriptive and correlation analyses were conducted for all variables, followed by an examination of the common method bias using Harman’s single-factor test. Then, the mediating role of expressive suppression and the moderating role of ego-identity in the relationships between collectivism and mental health, and expressive suppression and mental health were examined when controlling for age by using the Model 15 of PROCESS. To test statistical significance, 95% confidence intervals of the bias-corrected boot-strapped method based on 5000 samples were used. 

## 3. Results

### 3.1. Common Method Biases 

In order to avoid common methodological deviations, the Harman single-factor method was used for statistical control; the results show that there were 12 factors with a characteristic value greater than 1, and the first factor explained a variation of 14.62%, which was far less than the 40% critical value. Therefore, the influence of common method deviation on the results of this study can be excluded. 

### 3.2. Description and Correlation Analysis 

The means, standard deviation, and correlation analysis of each variable in this study are shown in Table 1. Collectivism was significantly negatively correlated with mental health and significantly positively correlated with expressive suppression. 

### 3.3. Test of Moderated Mediation

To test the hypothesized model, we used SPSS macro-PROCESS, which is able to test the moderated mediation in a single model and has been used by numerous researchers [38]. First, we calculated the effect of collectivism on mental health. Then, we added expressive suppression as a mediator. Last, we evaluated the moderating role of commitment, crisis, and exploration, respectively. After controlling for age, we examined the moderated mediating model by using Model 15 of PROCESS Macro. 

#### 3.3.1. Crisis as a Moderating Variable

When the mediating variable was included, the direct effect of collectivism on mental health was not significant. Collectivism had a positive association with expressive suppression (β = 0.17, *p* < 0.001). Expressive suppression had a significant negative association with mental health (β = −3.42, *p* < 0.01). In addition, the upper and lower bounds of the bootstrap 95% confidence interval of the mediating effect of expressive suppression did not contain zero, indicating that expressive suppression can play a mediating role in the link between collectivism and mental health.

Table 2 shows the results of the moderated mediation analysis. Interaction 1 between collectivism and crisis was negatively associated with mental health (β = −0.12, *p* < 0.05). Interaction 2 between expressive suppression and crisis was positively associated with mental health (β = 0.21, *p* < 0.01). In addition, the index of the moderated mediation model was 0.04 (95% CI = [0.01, 0.08]). That is, crisis plays a moderating role in the effect of collectivism on mental health, and the effect of expressive suppression on mental health.

Furthermore, we conducted a simple slope analysis and plotted the effect of collectivism on mental health separately with high or low crisis (Figure 2). The results show that, compared with the low-crisis group (b_simple_ = −0.20, *p* > 0.05), higher levels of collectivism were more strongly predictive of lower levels of mental health among the high-crisis group (b_simple_ = −0.73, *p* < 0.001). That is, high crisis may exacerbate the negative association between collectivism and mental health. Similarly, we also performed a simple slope analysis and plotted the effect of expressive suppression on mental health separately with high or low crisis (Figure 3). The results show that, compared with the high-crisis group (b_simple_ = 0.15, *p* > 0.05), higher levels of expressive suppression were more strongly predictive of lower levels of mental health among the low-crisis group (b_simple_ = −0.79, *p* < 0.01). That is, low crisis may exacerbate the negative association between expressive suppression and mental health. The above results suggest the moderated mediation model was supported.

#### 3.3.2. Commitment as a Moderating Variable

The interaction between collectivism and commitment was non-significantly associated with mental health (β = 0.06, *p* > 0.05). The interaction between expressive suppression and commitment was non-significantly associated with mental health (β = 0.07, *p* > 0.05). In addition, the index of the moderated mediation model was 0.01 (95% CI = [−0.01, 0.03]). That is, the moderated mediation model was not significant.

#### 3.3.3. Exploration as a Moderating Variable

When the mediating variable was included, the direct effect of collectivism on mental health was negatively significant (β = −3.26, *p* < 0.01). Collectivism had a positive association with expressive suppression (β = 0.17, *p* < 0.001). Expressive suppression had a significant negative association with mental health (β = −2.95, *p* < 0.05). In addition, the upper and lower bounds of the bootstrap 95% confidence interval of the mediating effect of expressive suppression did not contain zero, indicating that expressive suppression can play a mediating role in the link between collectivism and mental health.

For moderated mediation analysis, Interaction 1 between collectivism and exploration was positively associated with mental health (β = 0.19, *p* < 0.05). The interaction between expressive suppression and exploration was positively associated with mental health (β = 0.19, *p* < 0.05). However, the index of the moderated mediation model was not significant (Index = 0.03, 95% CI = [−0.00, 0.07]). That is, the moderated mediation model was not significant.

## 4. Discussion 

Based on emotion regulation and psychological development, this study explores the relationship between collectivism and mental health in the context of COVID-19. Moreover, this study provides a more in-depth discussion to improve mental health under a Public Health Emergency of International Concern (PHEIC). 

### 4.1. The Mediating Role of Expressive Suppression

The present study found that expressive suppression completely mediated the relationship between collectivism and mental health. Collectivism indirectly and negatively predicted mental health mainly by influencing people’s choice of emotion regulation and making them choose expressive suppression more often. That is, the more people choose expressive suppression in their emotion regulation, the more conscious suppression of emotion expression they engage in, which in turn produces more negative emotional experiences. In addition, the outcomes are similar to a previous meta-analysis [39]. It is particularly noteworthy that the original significant negative association between collectivism and mental health disappeared after the inclusion of expressive suppression as a mediating variable, which may indicate that the effect of collectivism on mental health is indirectly generated through expressive suppression.

Our finding highlights the important role of emotion expression to build mental health, especially in our collective culture. There is the risk of producing a negative effect on mental health due to expressive suppression, which highlights the importance of sharing bad feelings and looking on the bright side in daily life intervention. Cui et al. [40] indicated that, in recent years, Chinese people have started to express their intimate opinions and emotions with others. To encourage these behaviors, we can organize workshops in the community to help community members express and clear up their bad mood in a more conscious way.

### 4.2. The Moderating Role of Ego Identity between Expressive Suppression and Mental Health

The results reveal a moderating role of ego identity between expressive suppression and mental health. Chinese people have been using expressive suppression strategies [41], and findings suggest that expressive suppression is correlated negatively with positive indicators of mental health [22,42,43]. In addition, there is a strong relationship between ego identity and mental health. The establishment of ego identity for individuals is a kind of integration of oneself living a healthy life. This is because low ego identity people are unable to integrate their past, present, and future experiences, and are unable to achieve harmony in their physical, psychological, and social selves. Under such conditions, it leads to a sense of worthlessness and meaninglessness, resulting in many psychological problems. Some scientists followed adolescents over time and noted that the relationship between individuals’ identity formation and their mental health showed that, the higher the ego identity, the better the mental health [44]. In this study, we found that the role of crisis in ego identity was particularly important. Expressive suppression significantly and negatively predicted mental health in low ego identity people, whereas this predictive effect was absent in high ego identity people. It is suggested that crisis plays an important role in individuals who choose expressive suppression, and high crisis may reduce the negative effects of expressive suppression on mental health.

### 4.3. The Moderating Role of Ego Identity between Collectivism and Mental Health

The present study also found that crisis moderated the relationship between collectivism and mental health. Specifically, collectivism significantly negatively predicted mental health in high-crisis people, whereas this predictive effect was in absent low-crisis people. These outcomes are different from previous studies. In past research, collectivism can be a protective factor for mental health [5]. Liu et al. [45] found that the prevalence of self-reported psychological distress was higher in the UK (34.1%) than in the Shanghai sample (14.1%) during COVID-19. However, this study also suggests that the reason for the difference in results between the two countries may be related to the severity of COVID-19 at the time. According to the data of the World Health Organization [46], when the survey was completed in the UK (July 2020), there were 167,150 cumulative cases and 4729 new cases, but in China, at this time, there were 85,245 cumulative cases and five new cases. If people in the UK felt more vulnerable to the epidemic, this perhaps impacted self-reported stress levels.

As for Ebola, which is also a PHEIC event, in the context of Ebola in 2014, Kim et al. [47] investigated how collectivism can act as a buffer against psychological problems in a U.S. sample. They found that, when individuals with high collectivism felt higher perceived vulnerability, they have lower levels of xenophobia, which is a component of the psychological threat response. It is suggested that high collectivists have lower psychological threat perceptions or behavioral responses in the face of Ebola. In other words, collectivism increases the protection efficacy and collectivists perceive the collective as being able to keep them safe. Thus, collectivism buffers and abates xenophobia, resulting in a higher sense of psychological security and stability.

However, this outcome was not validated in this study. The possible reason is that China is a typical collectivist country, while the United States is an individualist country. The very nature of American collectivism is different from that of East Asian Confucian collectivism. The reason is referenced by Knyazev et al. [8], who found collectivism did not act as a protective factor against depression in Russia (a collectivist country). Another possibility might be related to the specific context and culture. The data for this study were collected in January 2021, when China was in the second outbreak of COVID-19, with five high-risk regions and 40 medium-risk regions across the country, and four large outbreaks in Hebei, Beijing, Liaoning, and Heilongjiang. A second massive outbreak enabled people to fall into fear again. Whether we can beat the epidemic and return to normal life is in doubt.

Moreover, collectivism deepens social ties among people and strong social ties motivate people to actively pay attention to external information and share their real-time situations, which increases individual and collective perceived risk [9]. Yang et al. [9] also indicated that the Chinese who were exposed to interpersonal communication and media would learn more about COVID-19 and increased the perceived risk. Then, perceived risk can decrease mental health [48].

## 5. Limitations

Theoretically, ego identity constitutes an aspect of optimal psychological functioning that drives individuals to have a healthier mental status [49]. However, the results of the present study found that collectivism is negatively associated with mental health in high-crisis people, i.e., high ego identity made the negative effect of collectivism on mental health more pronounced. This suggests that ego identity may not function well as a positive effect on mental health in the context of the typical collectivist culture and COVID-19 in China. However, the mechanisms are difficult to account for in this study, and the reasons for this situation and other factors can be further explored in the future.

Our study was based on self-reporting, due to the authenticity of self-reporting and social desirability, our data may not have high reliability over Cronbach’s alpha 0.9. For example, Cronbach’s alpha of the Identity Status Scale (ISS) was 0.672. Future studies may adopt multiple methods and more objective indicators to expand our findings.

In addition, there is no consensus on whether crises have an effect on mental health [29,30,31], but in this study, only crisis was found to have an effect on mental health among the three dimensions of ego identity. Future research could continue to explore the mechanisms of the effect between crisis and mental health.

## 6. Conclusions

The findings reveal that collectivism has a significant influence on mental health through expressive suppression. In the future, we can pay more attention to citizens’ emotion regulation. Moreover, the relationship between collectivism and mental health, and expressive suppression and mental health are moderated by ego identity. Especially, ego identity cannot play a positive role between collectivism and mental health. Future studies may explore the potential causes further.

## Figures and Tables

**Figure 1 ijerph-19-15570-f001:**
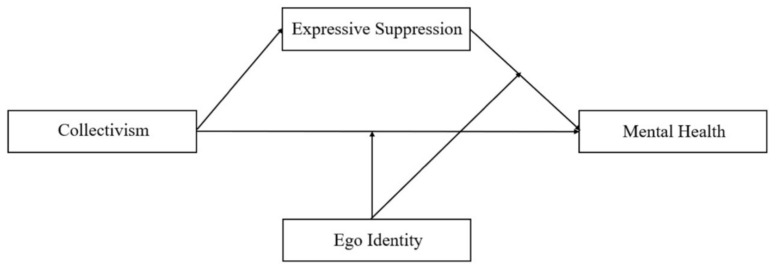
Schematic of moderated mediation model.

**Figure 2 ijerph-19-15570-f002:**
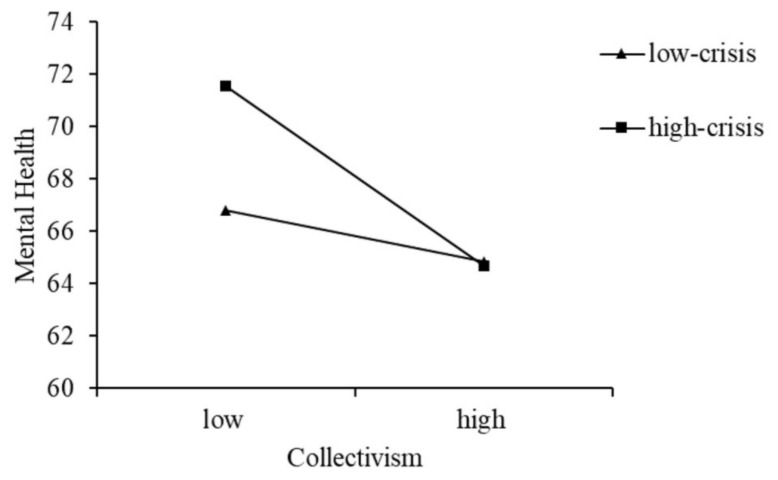
Moderating role of crisis in the relationship between collectivism and mental health.

**Figure 3 ijerph-19-15570-f003:**
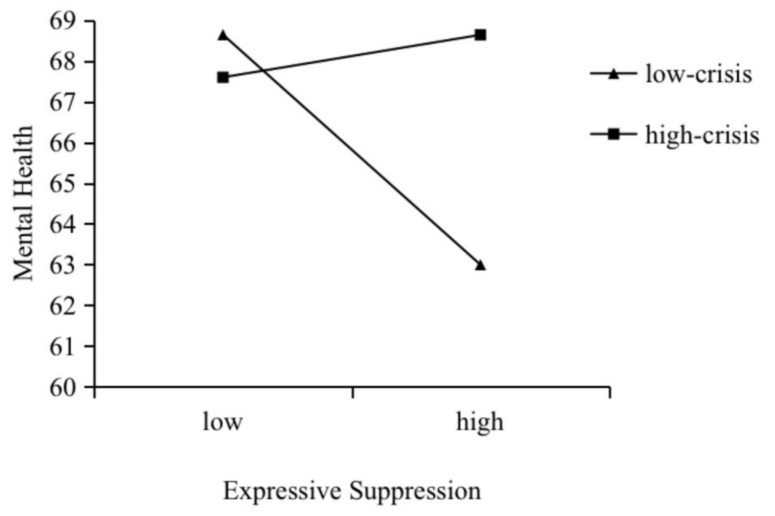
Moderating role of crisis in the relationship between expressive suppression and mental health.

**Table 1 ijerph-19-15570-t001:** Results of descriptive statistics and correlation analysis for each variable (corrected by Bonferroni Correction) (N = 459).

Variable	M	SD	1	2	3	4	5	6	7	8	9	10	11	12	13
Gender	1.63	0.48	1												
Age	21.25	6.58	0.04	1											
Mental health (revised)	66.88	13.01	−0.18 *	−0.07	1										
Fear (revised)	22.84	6.06	−0.27 ***	−0.04	0.91 ***	1									
Depression (revised)	20.49	3.125	−0.05	0.04	0.58 ***	0.45 ***	1								
Anxiety (revised)	14.02	4.47	−0.05	−0.14	0.79 ***	0.59 ***	0.19 **	1							
Anger (revised)	9.52	2.9	−0.08	−0.06	0.76 ***	0.58 ***	0.30 ***	0.55 ***	1						
Crisis	14.25	2.17	−0.04	−0.09	0.11	0.04	0.14	0.11	0.09	1					
Commitment	14.55	2.98	−0.09	0.13	0.13	0.12	0.38 ***	−0.07	0.01	0.31 ***	1				
Exploration	13.68	1.68	−0.02	−0.02	0.08	0.06	0.30 ***	−0.04	−0.03	0.32 ***	0.53 ***	1			
Collectivism	33.25	4.68	0.02	0.08	−0.20 **	−0.13	0.02	−0.28 ***	−0.21 ***	−0.01	0.21 ***	0.18 **	1		
Cognitive reappraisal	26.06	3.98	−0.04	0.04	−0.07	−0.06	0.16	−0.14	−0.14	0.22 ***	0.36 ***	0.32 ***	0.45 ***	1	
Expressive suppression	13.62	3.56	−0.20 **	0.04	−0.12	−0.02	−0.15	−0.14	−0.11	−0.06	0.02	−0.04	0.22 ***	0.24 ***	1

Note: All correlation analysis was corrected by Bonferroni Correction [36,37]; Gender: male = 1, female = 2; * *p* < 0.05, ** *p* < 0.01, *** *p* < 0.001.

**Table 2 ijerph-19-15570-t002:** Moderated mediating model test.

Variable	Outcome Variable: Expressive Suppression					Outcome Variable: Mental Health				
	β	se	t	LLCI	ULCI	β	se	t	LLCI	ULCI
Age	0.011	0.025	0.432	−0.038	0.059	−0.094	0.090	−1.045	−0.272	0.083
Collectivism	0.169	0.035	4.848 ***	0.100	0.237	1.256	0.818	1.535	−0.352	2.863
Expressive suppression						−3.418	1.145	−2.986 **	−5.668	−1.168
Crisis						1.602	1.870	0.857	−2.074	5.277
Interaction 1						−0.121	0.057	−2.137 *	−0.233	−0.010
Interaction 2						0.217	0.078	2.774 **	0.063	0.371
R2	0.050					0.077				
F	12.074 ***					6.262 ***				

Note: Interaction 1 is the product term of collectivism and crisis, and Interaction 2 is the product term of expressive suppression and crisis. * *p* < 0.05, ** *p* < 0.01, *** *p* < 0.001.

## Data Availability

The data presented in this study are available upon request from the corresponding author.

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
