# Peer review of "The Effect of Collectivism on Mental Health during COVID-19: A Moderated Mediation Model"

_ijerph, 2022, doi:10.3390/ijerph192315570_

Round 1

Reviewer 1 Report

The COVID-19 pandemic was an unprecedented, historic event that had a huge impact on the physical and mental health of societies. Research aimed at understanding this impact, especially in the area of ​​mental health, is both needed and valuable – not only for science (comprehension and prognosis) but also for more effective management in crisis situations. The article submitted for review tests the mechanisms of people's feelings, taking into account their complexity (mediation and moderation analyses). The article is interesting and written in accordance with the principles of how to present research results. I recommend the article for publication after making corrections in accordance with the following remarks:

-    43-45: This is illogical. Since collectivism was negatively associated with neuroticism, depression and deregulation of emotions, it would mean that it had a positive effect on health. Probably the authors mistakenly used the term ‘negatively associated’ instead of ‘positively associated’

-    The Identity Status Scale (ISS) - Cronbach’s alpha for the present sample was 0.672. This is a low result, below the conventional value of 0.7. I don't think it poses a threat to the reliability of the tool. But it should be mentioned.

-    No information is given as to whether the data used for statistical analyses (results for individual variables) were checked in terms of meeting the condition of normal distribution (in particular in terms of skewness, kurtosis and standard tests – e.g., the Smirnow-Kolmogorov-Smirnov or Shapiro-Wilk test) . Please indicate this and provide test results

-    With so many correlation analyses, it is worth applying a correction for multiple tests – e.g., the Holm-Bonferroni correction

-    238: The authors write that the direct effect of collectivism on mental health was negatively significant, whereas the given value of β = 0.17 is positive – please correct

- Different fonts are used (page 7) – this requires editorial correction

-   in the Limitations section, limitations should be described that are related to the method used, and not only to the interpretation of the results

Author Response

Dear Reviewer 1: Thank you so much for your precious comments. Please see the attachment.

Reviewer 2 Report

The authors of the article should be congratulated for choosing the topic of research, building an accurate, as shown by the obtained results, theoretical model referring to the considerations and research results included in the literature on the subject, and then verifying it. Similarly, the methodological part of the article was prepared communicatively and correctly.

However, to further increase the readability of the article to a greater extent, the previously formulated problems and/or research hypotheses, included in lines (58-59; 89-90), should be distinguished. According to the reviewer, they should be included in the beginning of the methodological part of the article (before line 125).

Despite the extensive part related to the discussion of the research results, the presented conclusions seem relatively modest and are purely cognitive, therefore the reviewer proposes to expand them to consider applications e.g. a proposal to organize workshops for Chinese youth, in a way that increases the repertoire moderating the expression of emotions and modifying them in a more conscious way.

Author Response

Dear Reviewer 2: Thank you so much for your precious comments. Please see the attachment.
